# Reducing Slip Risk: A Feasibility Study of Gait Training with Semi-Real-Time Feedback of Foot–Floor Contact Angle

**DOI:** 10.3390/s22103641

**Published:** 2022-05-10

**Authors:** Christina Zong-Hao Ma, Tian Bao, Christopher A. DiCesare, Isaac Harris, April Chambers, Peter B. Shull, Yong-Ping Zheng, Rakie Cham, Kathleen H. Sienko

**Affiliations:** 1Department of Mechanical Engineering, University of Michigan, Ann Arbor, MI 48109, USA; czh.ma@polyu.edu.hk (C.Z.-H.M.); baotian@umich.edu (T.B.); cdicesare@exponent.com (C.A.D.); isharris@umich.edu (I.H.); 2Department of Biomedical Engineering, The Hong Kong Polytechnic University, Hong Kong 999077, China; yongping.zheng@polyu.edu.hk; 3Department of Bioengineering, University of Pittsburgh, Pittsburgh, PA 15260, USA; ajchambers@pitt.edu (A.C.); rcham@pitt.edu (R.C.); 4Department of Health and Human Development, University of Pittsburgh, Pittsburgh, PA 15260, USA; 5Department of Mechanical Engineering, Shanghai Jiao Tong University, Shanghai 200240, China

**Keywords:** foot–floor contact angle, slip, feedback, gait training, inertial measurement unit

## Abstract

Slip-induced falls, responsible for approximately 40% of falls, can lead to severe injuries and in extreme cases, death. A large foot–floor contact angle (FFCA) during the heel-strike event has been associated with an increased risk of slip-induced falls. The goals of this feasibility study were to design and assess a method for detecting FFCA and providing cues to the user to generate a compensatory FFCA response during a future heel-strike event. The long-term goal of this research is to train gait in order to minimize the likelihood of a slip event due to a large FFCA. An inertial measurement unit (IMU) was used to estimate FFCA, and a speaker provided auditory semi-real-time feedback when the FFCA was outside of a 10–20 degree target range following a heel-strike event. In addition to training with the FFCA feedback during a 10-min treadmill training period, the healthy young participants completed pre- and post-training overground walking trials. Results showed that training with FFCA feedback increased FFCA events within the target range by 16% for “high-risk” walkers (i.e., participants that walked with more than 75% of their FFCAs outside the target range) both during feedback treadmill trials and post-training overground trials without feedback, supporting the feasibility of training FFCA using a semi-real-time FFCA feedback system.

## 1. Introduction

Slip-induced falls comprise up to 40% of falls among older adults and can cause significant injuries with lasting negative effects on daily activities [1]. Occupational slips lead to an absence of approximately 11 working days post-injury due to the subsequent injuries [2]. Slips can be caused by various factors including an improper contact surface between foot and ground and a large foot–floor contact angle (FFCA) at heel strike [3,4,5,6,7,8]. For example, prior studies have shown that a large FFCA (>20°) is associated with more frequent slips [3,6] and falls [4,5].

To prevent slips and falls, multiple interventions have been developed to improve the contact surface between foot and ground, such as non-slip socks and footwear modifications. However, the effectiveness of non-slip socks on slip-prevention remains unclear [9]. A recent systematic review reported that insoles (e.g., orthopedic, textile, or vibrating insoles) generally improve static and dynamic balance, but may not be able to address the environmental risk factor of a slippery floor [10]. Slip-resistant shoes can increase friction between footwear and ground surfaces, and reduce the severity of the slip [11,12]. Training on a movable floor has been shown to improve slip recovery [13], but findings have been limited to indoor settings, and many slips happen outdoors [2]. Another method shown to reduce slips and falls is walking with proactive gait adaptations that decrease both slip probability and slip severity. Beneficial proactive gait adaptations include shortened step length, increased cadence, and decreased FFCA [5,14].

Feedback systems coupled with inertial measurement units (IMUs) can provide cues based on the measured kinematics to improve balance [15,16,17,18,19,20,21,22,23,24,25,26] and gait [27,28,29,30,31]. For example, a feedback system with an IMU at the trunk has been demonstrated to reduce trunk tilt during walking [27]. Previous studies have also reported that IMUs measuring lower-limb kinematics paired with corresponding haptic or visual cues can correct excessive internal and external foot rotation during walking (i.e., foot progression angle) in healthy older adults [32] and increase insufficient knee-flexion angle during the swing phase of gait in a participant with cerebral palsy [33]. A proof-of-concept study supported the feasibility of foot-mounted IMUs to improve certain gait parameters including stance/swing time, stride length, and FFCA [34]; significant changes were observed for participants with spinal cord injury and older adults, but not for participants with stroke, when participants were provided with verbal feedback of foot angles [34]. However, this particular study did not examine full-body kinematics [34], which may be helpful to explain the inconsistent results among different participant groups. In summary, a limited number of studies have explored the potential effects of providing feedback about FFCA during walking, and among the studies that have investigated this topic, none have considered the effects of FFCA alterations on full-body kinematics.

This feasibility study aimed to address three research questions: (1) can participants use feedback to change their FFCA during treadmill walking; (2) do outcomes from treadmill gait training carry over to overground walking; and (3) how do the body kinematics change with FFCA feedback? We hypothesized that an IMU-based FFCA feedback system could be used to detect FFCA and prompt participants to adjust their FFCA during subsequent heel-strike events if it exceeded or fell below a target range of values. Such a system could be used as a real-time aid or a gait-training tool in future practice.

## 2. Materials and Methods

### 2.1. Participants

Ten healthy young adults (three females and seven males, mean age 22.0 ± 1.6 years, mean weight 66.7 ± 5.9 kg, and mean height 173.4 ± 7.0 cm) were enrolled in this study. All participants provided written informed consent prior to participation. Approval was granted from an Institutional Review Board at the University of Michigan (HUM00015990). Following the completion of the baseline assessment (session 1), only seven participants (three females and four males, aged 21.9 ± 1.9 years, weight 66.3 ± 6.9 kg, and height 172.0 ± 7.8 cm) met the FFCA-based criteria and were included in the data analysis. For participants who did not meet the criteria, data collection ceased after the baseline assessment.

### 2.2. Experimental Protocol

The protocol comprised four experimental sessions conducted in a laboratory setting: session 1—baseline assessment; session 2—gait training with verbal instructions prior to each walking trial to reduce FFCA during each subsequent trial; session 3—treadmill gait training with semi-real-time feedback; and session 4—post-training assessment after the FFCA-feedback training. All sessions except for the third session (treadmill gait training) included both overground and treadmill walking trials. Overground walking trials were included in this preliminary study to assess the extent to which carry-over effects from training with feedback while walking on a treadmill were observed. Following the completion of the baseline assessment, participants’ FFCAs were analyzed to determine the percentage of heel-strike events resulting in FFCAs within the target range of 10°–20°. Participants who had more than 75% of their FFCAs within the target range during the baseline session were excluded from the study on the basis that minimal opportunities existed to further adjust their FFCA with the addition of feedback.

After the baseline assessment, each participant was provided with summary information about the percentage of their FFCAs that fell within the desirable range. If participants met the inclusion criteria, they were provided with more detailed information about their FFCA values and instructions on how to adjust their FFCA to achieve values within the target range. Each participant was allowed to practice walking a few steps within the targeted FFCA range to familiarize themselves with the task.

For the overground walking trials performed as part of sessions 1, 2, and 4, participants completed five trials per session and were instructed to walk naturally along a 7 m walkway. For the treadmill walking trials performed as part of all four sessions, participants were instructed to walk on a level treadmill (TMX 22, Fuller Vision Inc., Groveport, OH, USA) for 2 mins [35] at a speed of 1.35 m/s [36]. Prior to the start of the first treadmill trial, participants were allotted 2 min to acclimate to walking on the treadmill. During the treadmill training session (session 2), participants performed gait training over four consecutive blocks of time, each lasting 4 mins [37]. Kinematic data were collected on participants throughout all four sessions with a passive optical motion tracking system and a custom IMU-based feedback system described below. Participants were required to rest for 3 min after experimental sessions 1, 2, and 4, and following completion of each 4 min training block within session 3 [37]. For the verbal instruction session (session 2), mean and range FFCA values in addition to the percentage of desirable FFCAs from the previous trial were verbally provided to participants, and participants were instructed to adjust their gait to achieve FFCAs within the desirable range before the start of each subsequent walking trial. For session 3, gait training with semi-real-time feedback was performed on the treadmill in order to enable many steps to be taken without stopping or turns within the field of view of the motion tracking system that captured kinematic data.

### 2.3. FFCA Training

The feedback system estimated FFCA at the heel-strike event for each step of the participant’s dominant foot and provided semi-real-time audio cues every two steps when the FFCA was outside of the desirable range via the built-in speaker of the laptop performing data analysis. Semi-real-time cues have been shown to be more effective than delivering continuous real-time reminders for other motion-training applications (e.g., changing knee-flexion/extension angles [38]). The feedback system consisted of (1) a commercially available IMU (MTW2-3A7G6, Xsens Technologies B.V., Enschede, The Netherlands; sampling frequency 120 Hz [39]) attached to the mid-foot by adhesive tape to measure foot kinematics along three axes; and (2) a laptop that analyzed the IMU data, computed the FFCA at heel strike, and provided the semi-real-time auditory cue after heel strike (Figure 1). The system detected heel strike (and computed FFCA) using a custom algorithm. The FFCA was defined as the pitch angle at heel strike, and the pitch angle was estimated as the angle between the foot and floor. During walking, the heel strike was detected in two phases. In the first phase, a foot-in-motion period was identified. A foot-in-motion period started when the heel raised from the ground and ended when the entire foot was on the ground. Specifically, when the pitch angle was less than 0° for at least three continuous time stamps (i.e., 0.025 s), the heel was considered to have risen from the ground. When the standard deviation of acceleration was less than 0.1 g for 50 continuous time stamps (i.e., 0.42 s) after the heel rise, the foot was considered to have been on the ground and the pitch angle was reset to 0° for the next stride. In the second phase, the heel strike was identified at the time stamp when the maximum acceleration was detected during the foot-in-motion period.

The FFCAs for two continuous steps of the dominant foot were averaged and then compared to the target range on a rolling basis. If the average FFCA value was greater than 20°, a high-pitch tone was provided to participants to cue them to slightly reduce their FFCA during their subsequent step, and if the average FFCA value was less than 10°, a low-pitch tone was provided to participants to cue them to increase their FFCA slightly during their subsequent step. Each participant was also told that the goal was to walk without receiving any feedback from the system during the training. Only direction information (not magnitude information) was provided to the participants. Although an IMU was placed on each foot, only the IMU placed on the dominant foot was used as the input for the FFCA feedback algorithm.

### 2.4. Three-Dimensional (3-D) Motion Collection and Processing

Participants were instrumented with 35 retroreflective markers, with a minimum of three tracking markers per body segment. Markers were affixed with double-sided tape to the sternum, clavicle, C7 and T10 vertebrae, right shoulder blade; and bilaterally on the acromio-clavicular joints, lateral epicondyle of the elbows, styloid processes of the radius and ulna, hands, anterior superior iliac spine, posterior superior iliac spine, anterior mid-thighs, lateral femoral condyles, tibial tubercles, lateral malleoli, the central forefeet (between the second and third metatarsals), and the heels. Four additional tracking markers were mounted on a headband worn by the participants. A 10-camera, high-speed, passive optical motion analysis system (Vicon, Oxford, UK) sampled at 100 Hz was used to capture the three-dimensional marker trajectory data from each participant. Ground reaction forces in Newtons were collected with two embedded force platforms (AMTI, Watertown, MA, USA) sampled at 1000 Hz that were synchronized with the motion capture system for the overground walking trials only. After marker placement, a static trial was conducted with the participant in a neutral position with foot direction and placement standardized to the laboratory’s coordinate system. Marker trajectories were filtered using a low-pass, fourth-order Butterworth filter with a cutoff frequency of 12 Hz. A six-degree-of-freedom skeletal model was applied to the filtered trajectories to determine the position and orientation of each segment at each time sample, and the model was scaled to each participant’s height and weight. Lower-extremity Cardan joint angles (hip, knee, and ankle) were calculated using Visual3D (C-Motion, Inc., Germantown, MD, USA), and the foot angle was computed in Visual3D as the absolute angle between the foot segment and the horizontal in order to compare to the FFCA computed from the IMU.

### 2.5. Outcome Measures and Data Analysis

#### 2.5.1. Comparison of FFCA across Experimental Sessions

The first and last steps of each overground walking trial and the first and last ten steps of each treadmill walking block were removed for data analysis to avoid the variations associated with the initiation and termination of gait. All post-processing was performed using Matlab (version R2016b, MathWorks, Natick, MA, USA). The percentage of steps with FFCAs within the target range, FFCA mean and standard deviation (SD), FFCA coefficient of variance (CV), cycle time, stride length, stride width, and speed during walking were calculated for each experimental session.

#### 2.5.2. Comparison of FFCA Computed Using IMU, 3D Motion Capture, and Force Platforms

To assess the validity of the semi-real-time feedback system to accurately detect FFCA, we compared the FFCA computed using the IMU data during the overground trials (defined as “IMU”) with a ground-truth measure, which we specified as the foot angle at heel strike as measured with the 3D motion capture system and detected using the embedded force platforms (i.e., ground reaction force; defined as “GRF”). Specifically, for those heel-strike events for which there was a clear heel strike, i.e., events when the heel of the foot made evident contact in the middle of the force platform, we compared the FFCA for that specific heel-strike event as determined using the IMU and the foot angle as measured using 3D motion capture at the point at which the heel strike was detected by the force platform using a threshold of 10 N. In addition, we applied the FFCA algorithm to the 3D motion capture data to compute FFCAs based on the motion capture data at each detected heel strike (defined as “Vicon”).

To compare the FFCA across all three conditions (“IMU”, “Vicon”, and “GRF”), the IMU data were down-sampled to match the frequency of the motion capture data and were time-synchronized using cross-correlation (i.e., using the *xcorr* function in Matlab).

#### 2.5.3. Joint Coordination Profile Analysis

To investigate the strategies used by participants to adjust their FFCA, a modified vector-coding technique was used to quantify inter-joint coordination for the hip–knee and knee–foot sagittal plane joint couplings during the swing phase prior to foot–floor contact [40,41]. Coordination was quantified for a given joint coupling by first time-normalizing the gait cycle preceding foot–floor contact to 101 data points (representing 0–100% of the cycle). Following this temporal normalization, a coupling angle was computed for each joint pairing, which indicated the relative angular motion within the coupling. Afterward, intra-participant coordination variation was quantified for each of the examined joint couplings by calculating the coupling angle variability (CAV) across each participant’s trials for a given condition (e.g., across the five trials for the baseline (session 1) and post-treadmill training overground trials (session 4, Figure 2) [42].

Finally, the average CAV values for each joint coupling during the swing phase of the gait cycle, defined as the final 40% of the time-normalized gait cycle, were computed. The CAV during the antecedent swing phase (i.e., ~40% prior to heel strike) was used to quantify the amount of inter-stride variability between two lower-extremity sagittal joint kinematic pairings (hip–knee & knee–foot). To interpret the CAV findings, a coordination profiling technique was employed to examine how the participants utilized feedback to modify their FFCA and to highlight differences between participants in a single-participant design [43,44].

### 2.6. Statistical Analysis

Statistical analyses were conducted to assess FFCA differences across the four experimental sessions, and FFCA differences across the three FFCA computation methods (i.e., “IMU”, “Vicon”, and “GRF”). The averages of (1) percentage of FFCA matching the desirable range of 10–20°, (2) FFCA values, (3) standard deviation (SD) of FFCA values, and (4) coefficient of variance (CV) of FFCA values were computed for each participant to conduct the statistical analysis. The Shapiro–Wilk test was used to verify data normality. For non-normally distributed data, the Friedman Test was performed to examine the existence of significant differences among the four experimental sessions using SPSS (version 22, IBM Corporation, Armonk, NY, USA). Post-hoc planned comparison tests with Bonferroni correction (Wilcoxon Signed Ranks Tests) were conducted to determine if significant differences existed between conditions of (1) treadmill: baseline vs. training with the FFCA feedback system; (2) treadmill: baseline vs. post-training with the FFCA feedback system; (3) treadmill: verbal instruction vs. training with the FFCA feedback system; and (4) overground: baseline vs. post-training with the FFCA feedback system. The Kruskal–Wallis H test was used to assess differences among the three FFCA computation methods, with Dunn’s post-hoc test for multiple comparisons used to assess differences among these methods. The level of significance was set at 0.05.

## 3. Results

### 3.1. Effects of Semi-Real-Time Feedback System on FFCA during Walking

The sensitivity and specificity of the IMU for detecting FFCAs between 10° to 20° and providing the correct feedback were 99.0% and 96.4% as compared to the FFCAs captured by Vicon in a pilot study involving five participants, respectively. An example of the distribution of FFCAs for different treadmill walking conditions for an illustrative participant is shown in Figure 3. The percentage of FFCAs within the desirable range (10–20°) was low during the baseline assessment, and increased while receiving feedback, with the training results retained for short periods of time after training with the feedback.

Table 1 reports the FFCA mean and SD values, and the spatial and temporal gait parameters computed across all participants. As shown in Table 1, the mean percentage of desirable FFCAs increased by 24.1% during feedback training (*p* < 0.05) with respect to the baseline treadmill values, and increased by 40.6% for the post-training session without feedback (*p* < 0.05) with respect to the baseline treadmill values. The baseline treadmill mean FFCA (9.9°) increased to 13.7° during training trials (*p* < 0.05) and to 13.0° during post-training trials (*p* < 0.05) compared to baseline treadmill trials. It is also interesting to note that the mean variability of FFCAs was smaller during treadmill training trials (*p* < 0.05) and post-training trials (*p* < 0.05) than during the baseline treadmill trials, and was also smaller during treadmill training trials compared to verbal-instruction treadmill trials (*p* < 0.05). No significant differences among FFCA parameters were found for overground walking trials, although the *p* values for the mean percentage of desirable FFCAs (*p* = 0.075) and reduction in mean values of the post-training FFCAs (*p* = 0.093) were approaching significance.

For the spatial and temporal gait parameters, both walking speed and stride length during treadmill walking with verbal instruction significantly decreased from the baseline treadmill walking trials (*p* < 0.05), but the percentage differences were not large (<5%). Meanwhile, the gait cycle time for overground walking trials with verbal instruction significantly increased by 6.6% from the baseline overground walking trials (*p* < 0.05). There was also a tendency that the gait cycle time for treadmill walking trials with feedback decreased from the baseline treadmill walking trials (*p* = 0.091). For the overground walking trials, the speed and stride length decreased when walking with verbal instruction compared to baseline walking trials (*p* = 0.063), while the gait cycle time after the training increased from the baseline trials (*p* = 0.063), although no statistical significance was reached.

### 3.2. Gait Kinematics and Coordination Changes Pre- and Post-Training

From baseline to post-training, participants primarily tended to reduce their FFCA by shortening their stride length, evidenced by a slightly reduced hip flexion at heel strike and a slight offset in the swing phase (Figure 4). The duration of the swing phase also decreased post-feedback, which may also help explain the shortened stride length.

The coordination profiling analysis revealed that the participants varied in how they coordinated their movements to reduce FFCA. Specifically, of the seven participants, three exhibited reduced CAV post-testing (9.7 to 2.8 for the hip–knee and 12.1 to 4.6 for the knee–foot, on average), while the remaining four exhibited increased CAV (5.2 to 10.0 for the hip–knee and 7.5 to 13.6 for the knee–foot, on average) (Figure 5). The changes in walking speed were also inconsistent among the seven participants. Specifically, five participants walked more slowly and the remining two walked more quickly in post-training overground walking trials after the treadmill training (Figure 6).

### 3.3. Comparison of FFCA Values Computed based on IMU, Vicon, and GRF

A total of 32 available FFCAs (with validation from the GRF-based measurement) out of 105 IMU-measured overground FFCAs (5 walking trials × 3 overground walking conditions × 7 participants) were identified and used for the computation and statistical analysis. Regarding the different FFCA computing methods, computing the FFCA based on GRF tended to underestimate the FFCA values in general (Figure 7). There was a significant main effect on FFCA among the three computing methods (Kruskal–Wallis Test; H(2) = 11.32, *p* = 0.004) (Figure 7B). The post-hoc analyses (Dunn’s Test) revealed that the FFCA was significantly larger when using the FFCA algorithm with both the Vicon (87.5% of overestimation, *p* = 0.039) and IMU (84.4% of overestimation, *p* = 0.004) data than when detecting FFCA using the GRF data. Meanwhile, the Vicon- and IMU-computed FFCAs were not significantly different from each other (*p* = 0.714).

## 4. Discussion

This preliminary study demonstrated the immediate, retained, and carried-over effects of training with a wireless semi-real-time FFCA feedback system on healthy young adults’ FFCAs during walking. The main findings of this study supported that: (1) healthy young individuals could use the feedback to change their FFCA during treadmill walking; (2) the training outcomes from treadmill gait training could be retained to some extent during the post-training treadmill walking trials, but were not carried over to the overground walking trials; and (3) in addition to consistently walking with slower speeds and shorter stride lengths when using the FFCA feedback system, participants appeared to employ different coordination strategies in order to reduce their FFCA during walking.

While previous studies found that non-slip socks [9] and slip-resistant shoes [11,12] did not significantly change the FFCA at heel strike, this study showed that participants walked with an increased percentage of FFCAs within the targeted range during the use of and after using the FFCA feedback system, indicating that participants could use this feedback system as a training device to adjust their FFCA and walking patterns. In this study, verbal instruction was also effective at reducing FFCA; however, verbal instruction had limitations at the lower range of the targeted FFCA, leading to an excessively small FFCA with a mean value of less than 10°. This finding may further suggest that additional objective real-time feedback is required.

Multiple previous studies have investigated the utility of IMUs coupled with feedback systems for training gait kinematics in various populations [27,28,29,30,31,33,34]. Additionally, several studies have investigated the ability of such systems to support retained training effects [33,34]. With respect to the immediate/real-time effects observed in this study, our findings are consistent with the findings of previously described feedback systems that have involved the use of IMUs placed on lower limbs coupled with feedback to provide cues regarding lower-extremity joint/segmental angles [27,28,29,30,31,33,34]. For example, Schließmann et al. performed a proof-of-concept study involving IMUs placed on the feet to train the foot–ground angle at heel strike in older adults, participants with stroke, and participants with spinal cord injury [34]. They observed real-time significant changes in participants with spinal cord injury and older adults, but not in the participants with stroke [34]. After a four-week training period, the positive effects achieved through training were not retained during the follow-up assessment [34]. Because kinematic data were not collected during this study, the strategies used by participants for whom changes were observed during the training portion of the session were unknown [34]. Other previous studies have reported a positive reduction in internal and external foot rotation during walking (foot progression angle) in healthy older adults [45] and young adults [46], and an increase in knee-flexion angle during the swing phase of gait for a participant with cerebral palsy [33] in response to real-time feedback.

Regarding the short-term effects of training with the FFCA feedback system, the resulting FFCA outcomes following treadmill training were retained to some extent during the post-training treadmill walking trials but were not carried over to the overground walking trials. Several factors may have contributed to this finding including the limited time spent on training (totaling 16 mins during a single day), different walking patterns between the treadmill and overground conditions [37], and/or the order of the sessions performed in the study (i.e., overground trials were always performed last). Previous studies have reported significant retained effects for gait training after multiple consecutive days of training (ranging from 3 to 18 days) with feedback delivered by either a device [34,47] or a physical therapist [48]. This study observed that the percentage of FFCAs within the targeted range increased from 26.4% to 43.3% (*p* = 0.075), and the mean values of the FFCA also decreased from 20.9° to 19.1° (*p* = 0.093) during the post-training overground walking trials as compared to the baseline, with the difference approaching significance.

While participants consistently walked with slower speeds and shorter stride lengths when using the FFCA feedback system, they appeared to employ different coordination strategies to reduce their FFCA during walking. Among the seven participants, four responded by distinctly increasing the CAV of the hip–knee and knee–foot couplings, while the remaining three demonstrated a different response (i.e., decreasing CAV). In the present study, CAV can be interpreted as characterizing how variable each joint’s sagittal plane angular motion is relative to the other in a given coupling during the swing phase as the foot prepares to make contact with the floor. More succinctly, this measure can characterize how constrained or flexible the joint coupling is in achieving the FFCA goal as trained with the feedback [42]. Although coordination profiling has not been examined with respect to FFCA, previous work found that head position feedback increased CAV in the lower extremities during running [49]; specifically, Lim and colleagues suggested that the increased lower-extremity CAV resulted in greater redundancy of referent lower-extremity configurations that allowed participants to maintain a stable head position while running. Similarly, it is possible that the four participants who increased CAV post-training achieved FFCAs within the target range by exploiting the inherent redundancy in the lower extremities (resulting in more variable configurations). Conversely, those that reduced their CAV appeared to have constrained the entire leg (resulting in less variable configurations) to keep the FFCA within the target range. However, due to the small sample size of this study, it is difficult to draw a conclusion on these interpretations, and a larger study should be conducted to investigate CAV responses further.

While no statistical difference was observed in FFCA computed based on IMU and Vicon data in this study, it is interesting to observe that the FFCA computed based on GRF data was consistently smaller than the FFCAs computed based on Vicon and IMU data. This difference may have been caused by the threshold value of the ground reaction force used to determine the gait event of a heel strike via the force platform (i.e., GRF ≥ 10 N). A smaller threshold value may have reduced the difference. Additionally, the cushion effect of the soft tissues at the heel may also help explain this observation; prior research has examined the time associated with the compression of the soft tissues during heel contact, which has been commonly represented as a small hump/peak (before the two large peaks) on the curve of GRF measured by the floor-mounted force platform in a gait cycle [50,51,52].

The primary limitations of this study were the small sample size and the use of a young, healthy population. Although the long-term target populations are older adults and individuals with an increased risk of slipping due to disease or disability, we opted to demonstrate proof of concept using a readily accessible population of participants. These factors may limit the broad application and generalization of this study. Future work should also explore the effectiveness of mitigating the severity of slip events on slippery surfaces following training with the FFCA feedback system.

The findings from this study have implications for slip-risk-reduction training and slip-risk monitoring. For example, a FFCA feedback system similar to the one used in this study could be used to train gait for potential slip prevention, either as a real-time aid or as a training tool that would be removed during activities of daily living. Given the previously established association with slip risk [3,4,5,6,7,8], FFCA could be monitored during activities of daily living with a modified version of the hardware and software used in this study (e.g., smartphone paired with single IMU and custom software) to predict when individuals are at an increased risk of slipping.

## 5. Conclusions

In summary, this pilot study supports the feasibility of gait training with semi-real-time feedback of the foot–floor contact angle in healthy young adults. However, participants walked with slower speeds and shorter stride lengths when receiving the semi-real-time feedback. Participants also employed different body-coordination strategies to reduce FFCA during walking. The immediate effect of retaining the change in FFCA following training on the treadmill suggests some carry over to overground walking conditions, which could have implications for safe gait-training applications. Following further development and evaluation, this type of FFCA feedback system could potentially be used as a real-time aid or a gait-training tool to reduce FFCA. Further studies are still needed to determine the effects on reducing risks of slips and falls among populations with higher fall risks.

## Figures and Tables

**Figure 1 sensors-22-03641-f001:**
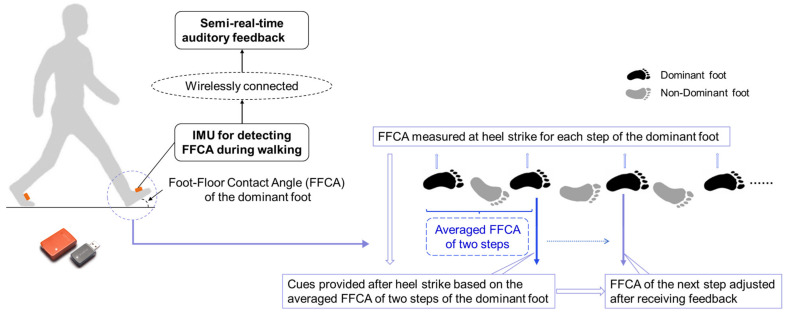
The semi-real-time system for providing feedback about the FFCA.

**Figure 2 sensors-22-03641-f002:**
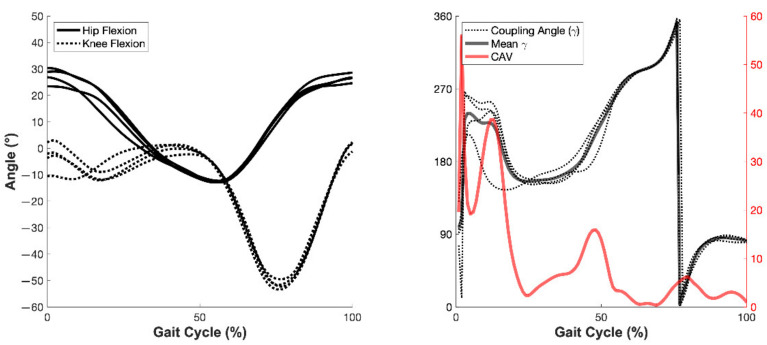
An illustration of the coupling-angle-variability (CAV) calculation process for the hip-flexion/knee-flexion joint coupling for a representative participant. The left figure shows hip and knee flexions across four gait trials; the right figure shows the coupling angle (γ) for the trials, the mean γ, and the CAV.

**Figure 3 sensors-22-03641-f003:**
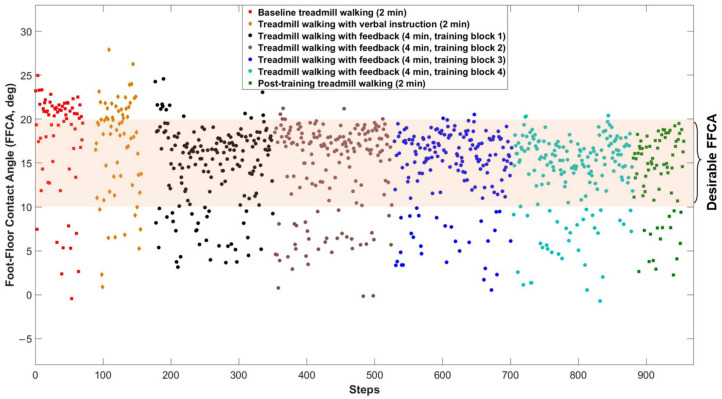
Example of the distribution of FFCAs for different treadmill walking conditions for a representative participant (*n* = 1). Each dot represents the FFCA of one step detected by the IMU.

**Figure 4 sensors-22-03641-f004:**
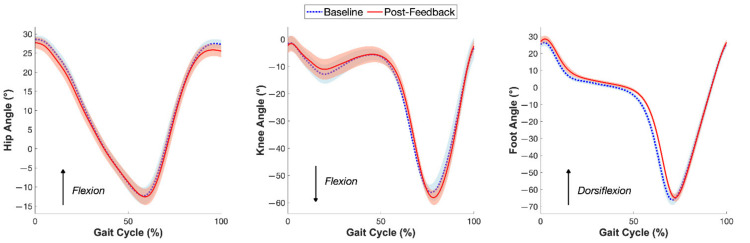
Mean gait cycle at baseline (blue) and post-feedback (red) of the hip joint, knee joint, and foot of the dominant leg (*n* = 7). The shaded regions indicate the 95% bootstrapped confidence interval around the mean for each curve.

**Figure 5 sensors-22-03641-f005:**
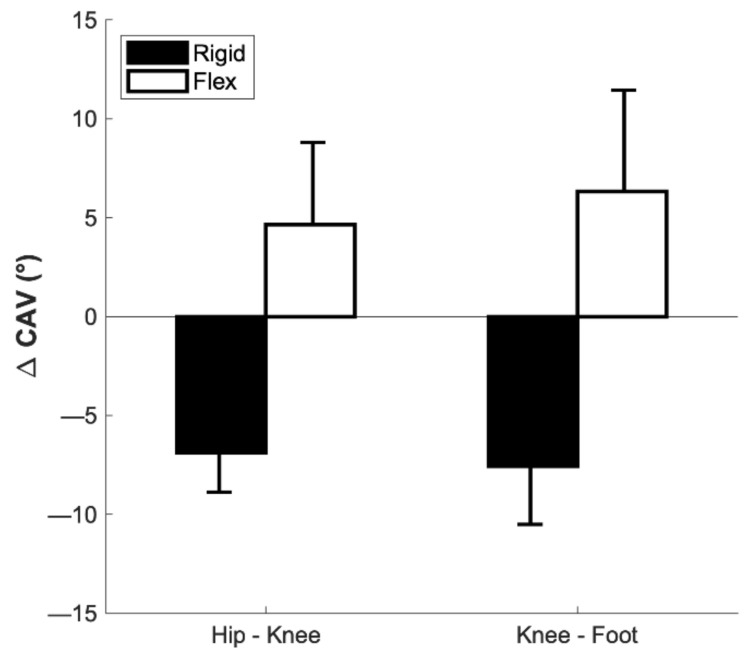
Average change in coupling angle variability (CAV) among the two joint pairings for the three participants who reduced their CAV baseline to post-feedback (Rigid) and the four participants who increased their CAV baseline to post-feedback (Flex) (*n* = 7). Error bars indicate the standard deviation.

**Figure 6 sensors-22-03641-f006:**
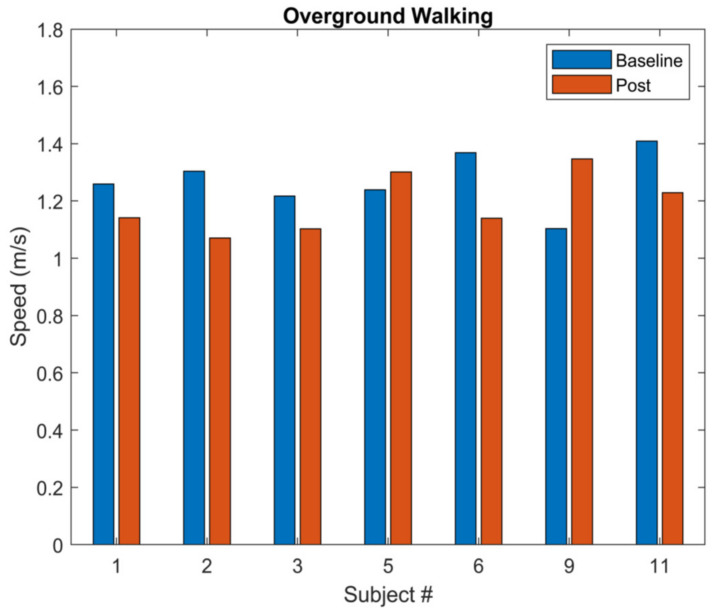
Overground walking speed of the seven participants for baseline and post-training walking trials (i.e., before and after the treadmill training) (*n* = 7).

**Figure 7 sensors-22-03641-f007:**
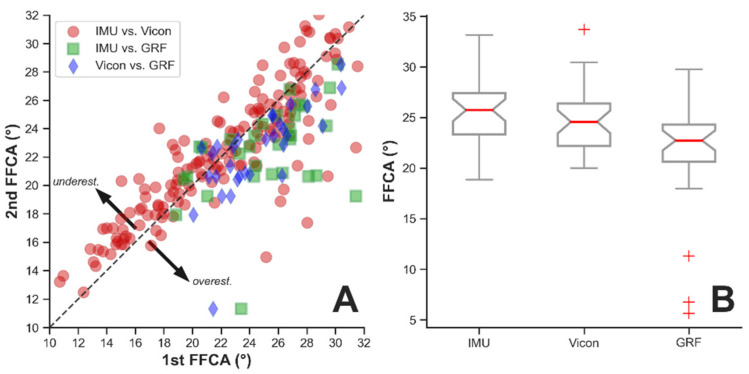
Comparison of FFCA computed based on IMU, Vicon and GRF: (**A**) Differences in FFCA computed based on IMU, Vicon and GRF. The “1st FFCA” is the first group listed in the legend entry (e.g., IMU in “IMU vs. Vicon”), while the “2nd FFCA” is the second group listed (e.g., Vicon in “IMU vs. Vicon”). For example, blue squares show how Vicon-based FFCA compares to GRF-based FFCA; (**B**) Boxplot showing the mean differences in FFCA computed based on IMU, Vicon and GRF. Red crosses indicate outliers.

**Table 1 sensors-22-03641-t001:** Comparison of the FFCA across all sessions (*n* = 7).

		Mean ± SD				*p*-Value				
		Baseline Session (S1)	Verbal Instruction Session (S2)	Feedback Training Session (S3)	Post-Training Session (S4)	Friedman Test (4 Sessions)	Planned Comparison (Wilcoxon Signed Ranks Test)			
							S1 vs. S2	S1 vs. S3	S1 vs. S4	S2 vs. S3
**Treadmill**	Percentage of desirable FFCA ^1^	53.9 ± 14.8%	45.9 ± 27.5%	66.9 ± 15.0%	75.8 ± 10.9%	<0.001 *	0.227	0.028 *	0.027 *	0.249
	FFCA (°)	9.9 ± 2.2	9.2 ± 3.4	13.7 ± 0.5	13.0 ± 1.4	<0.001 *	0.761	0.028 *	0.075	0.028 *
	SD of FFCA	6.5 ± 3.2	4.6 ± 1.2	5.4 ± 1.1	5.3 ± 2.6	0.009 *	0.311	0.463	0.046 *	0.027 *
	CV of FFCA	66.5% ± 27.0%	55.0% ± 20.5%	39.4% ± 8.3%	42.2% ± 25.2%	0.020 *	0.457	0.028 *	0.028 *	0.075
	Speed (m/s)	0.743 ± 0.013	0.740 ± 0.023	0.697 ± 0.085	0.744 ± 0.011	<0.001 *	0.046 *	0.176	0.866	0.499
	Gait cycle time (s)	1.278 ± 0.043	1.245 ± 0.064	1.267 ± 0.061	1.273 ± 0.070	<0.001 *	0.091	0.612	>0.999	0.866
	Stride length (m)	0.951 ± 0.038	0.922 ± 0.052	0.886 ± 0.121	0.948 ± 0.053	<0.001 *	0.046 *	0.176	0.866	0.499
	Stride width (m)	0.1408 ± 0.0379	0.1427 ± 0.0338	0.1430 ± 0.0382	0.1434 ± 0.0479	0.713	-	-	-	-
**Overground**	Percentage of desirable FFCA ^1^	26.4% ± 20.7	35.0% ± 22.7	-	43.3% ± 25.6	-	0.895	-	0.075	-
	FFCA (°)	20.9 ± 4.7	16.7 ± 4.9	-	19.1 ± 3.0	-	0.058	-	0.093	-
	SD of FFCA	5.7 ± 2.1	6.5 ± 0.9	-	6.9 ± 1.3	-	0.979	-	0.173	-
	CV of FFCA	29.0% ± 14.1	40.8% ± 10.1	-	36.0% ± 3.9%	-	0.809	-	0.249	-
	Speed (m/s)	1.272 ± 0.101	1.099 ± 0.147	-	1.190 ± 0.104	-	0.063	-	0.237	-
	Gait cycle time (s)	1.085 ± 0.045	1.157 ± 0.073	-	1.146 ± 0.057	-	0.043 *	-	0.063	-
	Stride length (m)	1.422 ± 0.116	1.302 ± 0.114	-	1.360 ± 0.093	-	0.063	-	0.237	-
	Stride width (m)	0.1434 ± 0.0495	0.1427 ± 0.0393	-	0.1374 ± 0.0386	-	-	-	-	-

FFCA: Foot–Floor Contact Angle; SD: Standard Deviation of FFCA; CV: Coefficient of Variance of FFCA; ^1^: Percentage of FFCA matching the desirable range of 10–20°; *: *p* < 0.05.

## Data Availability

The data presented in this study are available within the article.

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
