# Peer review of "Reducing Slip Risk: A Feasibility Study of Gait Training with Semi-Real-Time Feedback of Foot–Floor Contact Angle"

_sensors, 2022, doi:10.3390/s22103641_

Round 1

Reviewer 1 Report

I appreciated having the opportunity to review this paper. The aim of this study is to evaluate if the subjects can change FFCA during treadmill walking using semi-real-time feedback of the foot-floor contact angle, how their body's kinematics changed, and carry over effect after treadmill gait training. This article is well-written and well-organized. Below are some comments and suggestions:

1. It would be better to provide the exact p values if possible instead of "p<0.05", such as in lines 271-276, which could show more information regarding significance. If some of them are very small, it is better to show them as for example p<0.001.

2. In figure 4, adding a legend will deliver more clear information to the readers. It seems red line and dotted line denote the actual measurements, but what does red and blue shadow mean?

3. Figure 7 is very straightforward to show the comparison results. I am wondering it may be better to include a table with numbers of overestimation and underestimation, which will be helpful to highlight the difference.

Author Response

We would like to thank the reviewers for their constructive feedback and believe the manuscript is stronger as a result. Below we detail the changes to the manuscript made to address the reviewers’ comments/concerns.

Reviewer 1

I appreciated having the opportunity to review this paper. The aim of this study is to evaluate if the subjects can change FFCA during treadmill walking using semi-real-time feedback of the foot-floor contact angle, how their body's kinematics changed, and carry over effect after treadmill gait training. This article is well-written and well-organized. Below are some comments and suggestions:

  1. It would be better to provide the exact p values if possible instead of "p<0.05", such as in lines 271-276, which could show more information regarding significance. If some of them are very small, it is better to show them as for example p<0.001.

Below we include the exact p-values in the response letter; none are less than 0.001 or 0.01, therefore we propose to maintain the current formatting to be consistent with the reporting of p-values in our other publications.

  • “…Table 1 reports the mean and SD values of the FFCA and the spatial and temporal gait parameters computed across all participants. As shown in Table 1, the mean per-centage of desirable FFCA of baseline treadmill walking session increased 24.1% with feedback training (p=0.028), and increased 40.6% for the post-training session without feedback (p=0.027). The baseline treadmill mean FFCA (9.9°) increased to 13.7° during training (p=0.028) and to 13.0° during post-training (p=0.075) conditions. It is also interesting to note that the mean variability of FFCA was smaller during treadmill training (p=0.028) and post-training (p=0.028) conditions than during the treadmill baseline condition, and was also smaller during treadmill training than during the verbal instruction condition (p=0.027). No significant differences among FFCA parameters were found for overground walking conditions, although the p values for the mean percentage of desirable FFCA (p=0.075) and reduction in mean values of the FFCA (p=0.093) post-training were approaching significance.

  • For the spatial and temporal gait parameters, both walking speed (p=0.046) and stride length (p=0.046) during treadmill walking with verbal instruction decreased significantly from the baseline treadmill walking session, but the percentage differences were not large (<5%). Meanwhile, the gait cycle time for overground walking trials with verbal instruction increased 6.6% significantly from the baseline overground walking trials (p=0.043). There was also a tendency that the gait cycle time for tread-mill walking trials with feedback decreased from baseline treadmill walking session (p=0.091). For the overground walking sessions, the speed and stride length decreased when walking with verbal instruction from baseline (p=0.063), while the gait cycle time after the training increased from the baseline session (p=0.063), although no statistical significance was reached...”

  1. In figure 4, adding a legend will deliver more clear information to the readers. It seems red line and dotted line denote the actual measurements, but what does red and blue shadow mean?

As suggested, we have added a legend indicating that the blue and red lines represent the mean kinematic waveforms at baseline and post-feedback, respectively. We have also added a description to the caption indicating that the shaded regions around the mean represent the 95% bootstrapped confidence interval about the means on lines 318-321:

  •  

Figure 4. Mean gait cycle at baseline (blue) and post-feedback (red) of the hip joint, knee joint, and foot of the dominant leg (n=7). The shaded regions indicate the 95% bootstrapped confidence interval around the mean for each curve.

  1. Figure 7 is very straightforward to show the comparison results. I am wondering it may be better to include a table with numbers of overestimation and underestimation, which will be helpful to highlight the difference.

In order not to duplicate the data in Figure 7, we have added information about  overestimation  on lines 337-340:

  • “…that FFCA was significantly greater when using the FFCA algorithm with both the Vicon (87.5% of overestimation, p=0.039) and IMU (84.4% of overestimation, p=0.004) data than when detecting FFCA using the GRF data...”

Reviewer 2 Report

lines 27-31: please fill in the data: ”Citation: Lastname, F.; Lastname, F.; Lastname, F. Title.” (on page 1, left). Also, the year is 2022 (not 2021).

line 445: the Conclusions section can be expanded, with a short summary of the results of the paper and the impact of these results

lines 474-584: please check the guidelines for References, to comply with all the MDPI requirements for citing the references.

Overall, I think that the quality of the paper is very good and the contributions are clearly developed and presented. 

Author Response

We would like to thank the reviewers for their constructive feedback and believe the manuscript is stronger as a result. Below we detail the changes to the manuscript made to address the reviewers’ comments/concerns.

lines 27-31: please fill in the data: ”Citation: Lastname, F.; Lastname, F.; Lastname, F. Title.” (on page 1, left). Also, the year is 2022 (not 2021).

As suggested, we have updated the citation data on lines 4-7, stating: “…Ma, C.Z.-H.; Bao T.; DiCesare, C.A.; Harris, I.; Chambers, A.; Shull, P.; Zheng, Y.-P.; Cham, R.; Sienko, K.H. Reducing slip risk: A feasibility study of gait training with semi-real-time feedback of foot-floor contact angle. Sensors 2022, 21, x. https://doi.org/10.3390/xxxxx…”

The year is also updated as 2022.

line 445: the Conclusions section can be expanded, with a short summary of the results of the paper and the impact of these results

As suggested, the conclusions have been expanded on lines 453-462:

  • “In summary, this pilot study supports the feasibility of gait training with semi-real-time feedback of the foot-floor contact angle in healthy young adults. However, participants walked with slower speeds and shorter stride lengths when receiving the semi-real-time feedback. Participants also employed different body coordination strategies to reduce FFCA during walking. The immediate effect of retaining the change in FFCA following training on the treadmill suggests some carry over to over-ground walking conditions, which could have implications for safe gait training applications. Following further development and evaluation, this type of wearable feedback system could potentially be used as a real-time aid or a gait training tool to reduce FFCA. Further studies are still needed to determine the effects on reducing risks of slips and falls among populations with higher fall risks.”

lines 474-584: please check the guidelines for References, to comply with all the MDPI requirements for citing the references.

As suggested, the references have been updated according to the MDPI requirements on lines 485-598. 

Overall, I think that the quality of the paper is very good and the contributions are clearly developed and presented. 

Thank you for the constructive feedback.

Reviewer 3 Report

Brief summary: The present study deals with the design and assess of a method for detecting the foot-floor contact angle (FFCA) during gait by means of inertial measurement unit. During the experimental tests, a speaker provided acoustic feedback when the FFCA resulted outside a predefined range. The external feedback was used as training parameter during a 10 min treadmill training period. Results of experimental measures conducted after the training period showed that training increased FFCA events within the target range by 16% for “high-risk” walkers both during feedback treadmill trials and post-training overground trials without feedback.

Broad comments: The present work is generally well described and written. The topic well fits current researches and interest in gait analysis and risks of falls, the use of wearable devices and the objective estimation of gait parameters. Some small revisions are required:

-The introduction is well written and full of references to previous researches. The main scope of the study is reported at the end of the introduction.

-In the methodology section, all the subsections are well presented and all the information about instrumentation, protocol, subjects and data analysis are reported. Fig. 1 is clear and well describes the methods, even if two IMU were reported positioned on both feet. As you explained at line 134-135, it seems that only one unit has been used. Please, revised this detail. Moreover, I suggest changing the order of the first and the second subsections, in order to present the participant as first. A figure representing the positioning of markers (line 162) might be helpful. Figure 2, the xlabel is wrong because kinematics is expressed as % of gait, no stance.

- Fig 3: consider the possibility to fill the points with colour for a clearer representation.

-Table 1: the reported statistical tests are different from the explanation in the statistical analysis subsection. Please, revised. In addition, use the same number of significant digits for all the results (I suggested 2 significant digits).

Author Response

We would like to thank the reviewers for their constructive feedback and believe the manuscript is stronger as a result. Below we detail the changes to the manuscript made to address the reviewers’ comments/concerns.

Brief summary: The present study deals with the design and assess of a method for detecting the foot-floor contact angle (FFCA) during gait by means of inertial measurement unit. During the experimental tests, a speaker provided acoustic feedback when the FFCA resulted outside a predefined range. The external feedback was used as training parameter during a 10 min treadmill training period. Results of experimental measures conducted after the training period showed that training increased FFCA events within the target range by 16% for “high-risk” walkers both during feedback treadmill trials and post-training overground trials without feedback.

Broad comments: The present work is generally well described and written. The topic well fits current researches and interest in gait analysis and risks of falls, the use of wearable devices and the objective estimation of gait parameters. Some small revisions are required:

-The introduction is well written and full of references to previous researches. The main scope of the study is reported at the end of the introduction.

-In the methodology section, all the subsections are well presented and all the information about instrumentation, protocol, subjects and data analysis are reported. Fig. 1 is clear and well describes the methods, even if two IMU were reported positioned on both feet. As you explained at line 134-135, it seems that only one unit has been used. Please, revised this detail. Moreover, I suggest changing the order of the first and the second subsections, in order to present the participant as first. A figure representing the positioning of markers (line 162) might be helpful. Figure 2, the xlabel is wrong because kinematics is expressed as % of gait, no stance.

As suggested, we have added more information about the system and experimental protocol on lines 161-164:

  • “Although an IMU was placed on each foot, only the IMU placed on the dominant foot was used as the input for the FFCA feedback mechanism.”

An illustration of the Plug-in Gait marker placement can be freely accessed from the system manual online at the following link: http://www.idmil.org/mocap/Plug-in-Gait+Marker+Placement.pdf.

Because this information is accessible online, we have opted not to replicate the figure in the manuscript. 

As suggested, we have changed the order of the first and the second subsections (lines 81-128):

  • “2.1. Participants

Ten healthy young adults ….

2.2. Experimental Protocol

The protocol comprised four experimental sessions conducted …”

As suggested, we have changed the x-axis label of Figure 2 to “Gait Cycle (%)” on line 230:

Figure 2. An illustration of the coupling angle variability (CAV) calculation process for the hip flexion/knee flexion joint coupling for a representative participant. The left figure shows hip and knee flexion across four gait trials; the right figure shows the coupling angle (γ) for the trials, the mean γ, and the CAV.

- Fig 3: consider the possibility to fill the points with colour for a clearer representation.

As suggested, the Figure 3 has been revised by filling the points with color for a clearer representation on line 270:

Figure 3. Example of the distribution of FFCAs for different treadmill walking conditions for a representative participant (n=1). Each dot represents the FFCA of one step detected by the IMU in the feedback system.

-Table 1: the reported statistical tests are different from the explanation in the statistical analysis subsection. Please, revised. In addition, use the same number of significant digits for all the results (I suggested 2 significant digits).

As suggested, the descriptions in the statistical analysis subsection have been revised on line 258-262:

  • “…The Shapiro-Wilk test was used to verify data normality. For non-normally distributed data, the Friedman Test was performed to … Wilcoxon Signed Ranks Tests) were conducted to …”

We have also updated the manuscript to include three significant digits for all results.
